# Marker Assisted Selection of Malic-Consuming *Saccharomyces cerevisiae* Strains for Winemaking. Efficiency and Limits of a QTL’s Driven Breeding Program

**DOI:** 10.3390/jof7040304

**Published:** 2021-04-15

**Authors:** Charlotte Vion, Emilien Peltier, Margaux Bernard, Maitena Muro, Philippe Marullo

**Affiliations:** 1Unité de Recherche Œnologie EA 4577, USC 1366 INRAe, Bordeaux INP, ISVV, Université de Bordeaux, 33882 Villenave d’Ornon, France; charlotte.vion@u-bordeaux.fr (C.V.); epeltier@unistra.fr (E.P.); margaux.bernard@u-bordeaux.fr (M.B.); maitena.muro@u-bordeaux.fr (M.M.); 2Biolaffort, 33000 Bordeaux, France; 3CNRS, GMGM UMR 7156, Université de Strasbourg, 67000 Strasbourg, France

**Keywords:** wine yeast, malic acid, pH, breeding, malolactic fermentation, marker assisted selection

## Abstract

Natural *Saccharomyces cerevisiae* yeast strains exhibit very large genotypic and phenotypic diversity. Breeding programs that take advantage of this characteristic are widely used for selecting starters for wine industry, especially in the recent years when winemakers need to adapt their production to climate change. The aim of this work was to evaluate a marker assisted selection (MAS) program to improve malic acid consumption capacity of *Saccharomyces cerevisiae* in grape juice. Optimal individuals of two unrelated F1-hybrids were crossed to get a new genetic background carrying many “malic consumer” loci. Then, eleven quantitative trait loci (QTLs) already identified were used for implementing the MAS breeding program. By this method, extreme individuals able to consume more than 70% of malic acid in grape juice were selected. These individuals were tested in different enological matrixes and compared to their original parental strains. They greatly reduced the malic acid content at the end of alcoholic fermentation, they appeared to be robust to the environment, and they accelerated the ongoing of malolactic fermentations by *Oenococcus oeni*. This study illustrates how MAS can be efficiently used for selecting industrial *Saccharomyces cerevisiae* strains with outlier properties for winemaking.

## 1. Introduction

The yeast *Saccharomyces cerevisiae* is involved in many biotechnological processes including bioethanol, brewery, bakery and wine making [1]. This species displays a wide genetic diversity that generates a high phenotypic variability for many traits of interest. In the context of wine fermentation, the selection of industrial starters is particularly relevant since yeast modulates the final quality of the product [2,3]. Indeed, numerous studies have demonstrated that commercial *S. cerevisiae* strains are different regarding their ability to complete the alcoholic fermentation, their fermentation kinetics and their contribution to the final composition of volatile and non-volatile compounds [4,5,6,7,8,9].

The wine industry has developed basic genetic selection for making the best of phenotypic variability and proposing optimal starters. Indeed, clonal selection [10] and yeast breeding programs [9,11,12] are nowadays widely used for developing new wine starters; for documented reviews see [2,13,14]. Beside this applicative aspect, the genetic basis of trait variability has been studied for several decades by forward [15,16,17,18,19,20,21,22,23,24,25,26,27] and reverse [5,28,29,30] genetic approaches allowing the detection of numerous genes and allelic forms that possibly impact phenotypes of interest. Recently, we reviewed the phenotypic impact of 153 genes controlling traits of industrial interest [31]. The identification of such allelic variations and the steady decreasing cost of genotyping methods paved the avenue for combining marker assisted selection (MAS) with classical breeding (cross and segregation). Despite its very efficient use in plant selection [32] and animal breeding [33], MAS has been poorly used in order to optimize fungi. Indeed, only few authors implemented this approach for accelerating breeding programs [34,35,36,37,38]. Generally, alleles used had a high penetrance and controlled a large part of phenotypic variation. However, most of allelic variations have a low penetrance and a small effect on the phenotype [39]. This is particularly true when MAS is implemented in outbred backgrounds [40] due to modifier alleles that drastically reduce QTL expressivity. The aim of this work is to evaluate MAS efficiency for improving a complex quantitative trait in a four-parental genetic background which represents an applied case of yeast breeding program. 

We focused our investigation on the L-malic acid consumption by *S. cerevisiae* during the alcoholic fermentation, that is a highly variable trait among wine strains (Peltier 2018). The final concentration of this organic acid, originally present in grape, strongly contributes to the acidic pH of wine, which has a significant impact on its organoleptic qualities. Therefore, the control of its consumption is important for the choice of winemaking routes. *S. cerevisiae* is unable to consume all the L-malic acid of grape juice due to the lack of an active transporter [41]. Indeed, *S. cerevisiae* strains consuming more than 50% of total L-malic present in a grape juice were never reported [42,43]. During the winemaking process, malic acid can be transformed in L-lactic acid by wine-specific lactic bacteria *Oenococcus oeni* [44]. This biotransformation step, called malolactic fermentation (MLF), mostly occurs in red wines after or during the alcoholic fermentation. A part of L-malic acid can be also consumed by the malo-ethanolic fermentation (MEF) as well as by the fumarase activity [42,45].

Recently, the genetic determinism of malic acid consumption by *Saccharomyces cerevisiae* has been partially unraveled [46]. Numerous quantitative trait loci (QTL) related to the malic acid consumption were precisely mapped in the SBxGN background. Additional loci controlling the consumption of malic acid in *S. cerevisiae* have been also reported [47]. The functional exploration of these QTLs reveals the role of seven genes (*MAE1*, *PMA1*, *PYC2*, *PNC1*, *YBL036c, PTC5,* and *SDH2*) by reciprocal hemizygosity assay [46,47]. In this present work, a large-scale breeding program was implemented for addressing how much, the rational use of MAS can be implemented for boosting malic acid consumption during the wine alcoholic fermentation. First, optimal individuals of two unrelated F1-hybrids were crossed in order to get a new genetic background carrying many “malic consumer” loci. Eleven unlinked QTLs previously identified were used for implementing a MAS program. By this method, optimal individuals consuming more than 70% of the malic acid of grape juice were obtained. Such individuals were tested in different enological matrixes; they lower the malic acid content at the end of the alcoholic fermentation and accelerate the ongoing of inoculated malolactic fermentations. 

## 2. Materials and Methods

### 2.1. Yeast Strain Used and Culture Methods

Yeast (*Saccharomyces cerevisiae*) were propagated on YPD 2% (1% peptone, 1% yeast extract, 2% glucose) at 28 °C in both liquid and plate cultures (2% agar). Long term storage at −80 °C was achieved by adding one volume of glycerol to YPD 2% overnight cultures. The main yeast strains used were described in the Table 1 and are derived from the two F1-hybrids (M2xF15 and SBxGN) used in previous QTL mapping studies [26,48]. All the strains are homozygous for the *HO* locus and are therefore diploids. The strains GS-28b and GS-41b are meiotic spore clones of SBxGN while FM-8d is a meiotic spore clone of M2xF15. These three strains were used as allele donors in a breeding program aiming to fix in few individuals most of the alleles promoting the consumption of malic acid during the alcoholic fermentation. The pedigree of the strains generated is shown in Table 1. 

### 2.2. Spore Mating and Purification

Since all the strains are homothallic, F1-hybrids (FMGS-1 and FMGS-2) were obtained by spore to spore mating on YPD 2% using a routine method previously described [9]. After few hours, zygotes (trilobed shaped) were observed and manually isolated; their heterozygous genotype was verified by Mass Array genotyping (see below). Spore purification was carried out by using the hydrophobic properties of the spore cell wall. Each F1-hybrids was sporulated by cultivating 10^9^ cells in 5-mL of potassium acetate 1% (ACK) during three days at 24 °C. This high-density spore culture was digested with cytohelicase (10 mg/mL) for 2 h at 28 °C under permanent shaking. Spore preparation was then washed twice (de ionized water) and the pellet was recovered in 100 µL in one centrifuge tube (1.5 mL). The suspension was vigorously vortexed for 2 min and the suspension was removed by pipetting. The remaining mixture of spores and cells was purified by adding/removing 1 mL of de ionized water two times. The remaining spores were unstacked from the tube wall by adding 1 mL of sterile Nonidet P40 (0.01%). The purification rate obtained ranged between 85 and 95% of spore clones. The purified spores were then sonicated (30 min), incubated 2 h on YPD (28 °C) and plated on several Petri dishes in order to obtained more than 1000 isolated colonies per preparation.

### 2.3. Cell Culture and DNA Extraction in Microplates

Each single colony picked up from YPD-agar plate was cultivated in 96-Well microplate (coated with gas-permeable sheets (BREATHseal^TM^, Greiner Bio-one, Courtaboeuf, France) allowing CO_2_ production. Genomic DNA was extracted by using a Li-Ac SDS protocol previously adapted for microplate handling [49]. Broadly, 5 × 10^6^ cells (200 µL of overnight cultures) were centrifuged and incubated with 50 µL of 200 mM LiAc/1% SDS at 70 °C for 5 min. Genomic DNA was extracted by mixing cell lysates with 150 µL of pure ethanol and vortexed for 15 s. After a brief spin (2 min, 4000 rpm) the pellet was washed with 70% ethanol and was the solubilized in 200 µL of milliQ water at 60 °C for 5 min. Some liquid handling steps were assisted by the robotic platform NxP (Beckman Coulter, Villepinte, France).

### 2.4. Mass ARRAY Genotyping

Genotyping was mostly carried out by using Mass ARRAY technology (Agena Bioscience, San Diego, CA, USA), which allows large multiplex SNP detection as previously described [50]. Less than 15 ng of DNA were used for genotyping the strain. The Online Resource 1 lists the identity of wanted SNP that possibly enhance the malic acid consumption according to a wide QTL mapping study recently published. SNPs were encoded according to the international nomenclature [51]. The genotypes of each parental strain founder (M2, F15, SB, GN) and the sequence of primers used were also provided. Primers were designed using the tool MassARRAY^®^ Assay Design (v4.0.0.2) in order to amplify short fragments (<115 bases) with a molecular weight differences ranging between 16 and 80 Da. By this method, 11 bi allelic loci were simultaneously tracked corresponding to the following QTLs (*II_661*, *IV_31*, *IV_360*, *IV_414*, *VII_427*, *VII_480*, *XI_382*, *XI_631*, *XII_53*, *XV_491*, *XV_1052*) which control malic acid consumption in the SBxGN background. The SNP calling was carried out by the MassARRAY^®^ System suite with an average SNP calling of 94.3%. Control strains (SB, GN, F15, M2) were genotyped several times giving in any case the identical SNP calling. All clones showing a fully homozygous genotype and 10 or more loci assigned were included in the analysis, which corresponds to 593 and 797 clones for FMGS-1 and FMGS-2, respectively.

### 2.5. Alcoholic Fermentation Assays

#### 2.5.1. Grape Must

The grape musts Merlot 2015 (M15) and Sauvignon Blanc 2016 (SB16) used were provided by Vignobles Ducourt (Ladaux, France) and stored at −20 °C. Grape musts were sterilized by membrane filtration (nitrate cellulose 0.45 µm, Millipore, France). In order to test the phenotypic robustness of some relevant strains, these musts were spiked with different amounts of L-malic acid and nitrogen source (Appendix A). The mixture of amino acid used mimics the average composition of organic sources in a grape wine as previously described [12]. The composition of the different grape musts in fermenting sugar, nitrogen sources and malic acid content as well their pH are listed in Appendix A.

#### 2.5.2. Alcoholic Fermentation Monitoring

Small-volume alcoholic fermentations were implemented in screwed vials fermentations according to the general procedure described in [52]. Rapidly, 20 mL-screwed vials (Thermo Fisher Scientific, Bordeaux, France) were filled with 12 mL of filtered grape must and were tightly closed with screw cap-magnetic (Agilent Technologies, hdsp cap 18 mm PTFE/sil 100 pk, Les Ulis, France) perforated by hypodermic needles (G26–0.45 × 13 mm, Terumo, Shibuya, Tokyo, Japan) for allowing the CO_2_ release. Vessel was inoculated by 2.10^6^ viable cell.mL^−1^ precultured in liquid YPD for 24 h. Cellular concentration and viability was estimated by flow cytometry using a Cell Lab Quanta apparatus (Beckman Coulter, Villepinte, France) as described by Zimmer et al. (2014). Fermentation took place at 24 or 28 °C in shaken vials by using an orbital shaker (SSL1, Stuart, Vernon Hills, IL, USA) at 175 rpm. Fermentation kinetics were estimated by monitoring manually (2–3 times per day) the weight loss caused by CO_2_ release using a precision balance with automatic weight recording (LabX system, Mettler Toledo, Viroflay, France). The amount of CO_2_ released according to time was modeled by local polynomial regression fit [52] allowing the estimation of the time necessary to reach the maximum CO_2_ produced. 

### 2.6. Malolactic Fermentations Monitoring

At the end of the alcoholic fermentation, vials were placed for 24 h at 4 °C for settling yeast lees. Sedimented part were then solidified by soaking vials in a thin layer of liquid nitrogen. Thus, yeasts were eliminated in order to avoid lees impact on MLF and to reproduce enological-like conditions. Ten mL of wine were transferred into 10 mL cylindrical vials (Thermo Fisher Scientific, Bordeaux, France, ref: 11981523), closed with screw caps (Agilent Technologies, hdsp cap 18 mm PTFE/sil 100 pk, Les Ulis, France). The wines were inoculated with Lactoenos^®^ B7 Direct rehydrated lactic bacteria at a concentration of 10 mg·L^−1^, as recommended by the supplier (Laffort, Bordeaux, France). Fermentation samples were taken every 2 to 5 days for measuring L-malic acid decrease as a proxy of the MLF progress. The MLF ongoing was fitted by a 3 parameters logistic function using the R package (*drm* function of *drc* package), four kinetics parameters were extracted: t_MLF_lag, and t_MLFend, correspond to the time to initiate and to achieve the MLF, respectively. t_MLF50 corresponds to the time for consuming 50% of malic acid, and rmax_MLF is the maximal malic acid consumption rate (g·L^−1^·h^−1^). Data fit and MLF parameters are illustrated in the Appendix A.

### 2.7. Enzymatic Assay of Wine 

At the end of the alcoholic fermentation, a sample volume of 800 μL was manually transferred in Micronics tubes (Novazine, Lyon, France, ref: MP32033L) and stored at −20 °C. The concentrations of the following organic metabolites were measured: acetic acid, glycerol, malic acid, pyruvate, acetaldehyde and total SO_2_ using the respective enzymatic kits: K-ACETGK, K-GCROLGK, K-LMAL-116A, K-PYRUV, K-ACHYD, K-TSULPH, (Megazyme, Bray, Ireland) following the instructions of the manufacturer. Glucose and fructose were assayed by using the enzymatic method described by Stitt et al. [53]. All the enzymatic assays were performed by a robotic platform using the Bordeaux metabolomics facilities (http://metabolome.cgfb.u-bordeaux.fr/). Only the strains able to complete the fermentations (<1.5 g·L^−1^ of residual sugars) were retained. 

### 2.8. Use of Previous Phenotypic Datasets

Two previously published datasets were used for estimating the heritability and the efficiency of this marker assisted selection program. Fermentation phenotypes were measured in the same conditions and in the same grape musts allowing to compare the data. Malic Acid Consumption (*MAC*) was the ratio of malic acid consumed ([L-malic acid]_initial_-[L-malic acid]_final_)/[L-malic acid]_initial_ expressed in %. The *MAC* values of the 35 enological strains and the 94 progenies of SBxGN hybrid were extracted form Peltier et al. (2018a) [46] and Peltier et al. (2020) [52], respectively. The *MAC* values of M2xF15 offspring were measured in the same conditions described by Peltier et al. (2018b) [48] but are only released in this present work. 

### 2.9. Statistical Analyses 

All the statistical and graphical analyses were carried out using R software [54]. Plots were computed using the *ggplot2* package and analyses of variance were carried out using the *car* package. Four linear models were applied: LM1 investigated the effect of QTL numbers on *MAC* according to the formula:(1)$$y_{i}=m+EA_{i}+\epsilon_{i},$$where *y* is the MAC value for a strain carrying *i* Enhancers Alleles (*i* = 1…11) and $\epsilon_{i}$ the residual, *m* is the overall mean of the 154 FMGSs progenies phenotyped. The LM2 model estimated the effects of the following factors must, strain, temperature, nitrogen, and malic acid as well as the first order interaction of all the factors on MAC value according to the formula: (2)yijklm=m+straini+mustj+tempk+nitrol+malicm+interijklm^2+ϵijkmn,where $y_{ijklm}$ is the *MAC* value for a strain *i* (*i* = 1…6) fermented in a must *j* (*j* = 1,2) at the temperature *k* (*k* = 1,2) and containing the nitrogen level *l* (*l* = 1,2) and the malic acid level *m* (*m* = 1,2,3). Levels taken by each factor are given in the Online Resource 2. The term  interijklm^2 represents the first order interactions for each factor and $\epsilon_{ijklm}$ the residual, *m* is the overall mean of the 84 fermentations carried out in triplicate in an equilibrated fractional design. This model was refined with the models LM3 and LM4 allowing the estimation of each factors in the SB16 and M15 must, respectively. 

The analysis of variance of models LM2, 3 and 4 allows the estimation of the primary effect of the strain, the grape must, the temperature, the nitrogen and the malic acid concentration as well as their primary interaction effects. The normal distribution of residues as well as the homoscedasticity of the variances were tested by Shapiro test and Levene test (*car* package). 

## 3. Results

### 3.1. A Basic Breeding Strategy for Improving Malic Acid Consumption

The main goal of this study consists to significantly increase the *S. cerevisiae* consumption of malic acid during alcoholic fermentation. Previously, it has been reported that wine strains were unable to consume more than 50% of the malic acid grape juice in controlled conditions. This overall feature was previously confirmed in our laboratory by measuring the residual malic acid concentration at the end of the alcoholic fermentation for 35 strains in 5 grape juices [52]. Since the initial amount of malic acid varied according to the grape juice, we computed the percentage of malic acid consumed (*MAC*) for comparing strains within multiple environmental conditions. On average, *MAC* ranged between 30.0 and 53.0% within a population of 31 commercial starters plus the four founder strains of this study (SB, GN, M2, F15), that are haploid derivative from commercial starters. Strong differences between grape musts were found (Appendix A). The continuous distribution of this trait was illustrated for a red grape juice Merlot 2015 in the Figure 1a. Strains SB and GN had extreme phenotypes, metabolizing between 40.0% and 3.5% of malic acid, respectively. Strains M2 and F15 had an intermediate phenotype, consuming up to 20% of malic acid in the same conditions. This analysis demonstrated that MAC is a quantitative trait with important variation among commercial starters and therefore we decided to took advantage of this natural variation by initiating a selection program with a breeding approach.

We decided to use two previously generated mapping populations as starting material. These populations originated from two hybrids, resulting from the mating of SB with GN (SBxGN) and M2 with F15 (M2xF15) that were sporulated to generate 94 and 96 segregants, respectively. The rational being that meiosis segregation would reveal transgressive phenotypes. SBxGN offspring was phenotyped for MAC in a previous study [48] and the phenotypic characterization of M2xF15 offspring was achieved in the same conditions but the *MAC* values are only released in this present work (Appendix A). The fermentation conditions applied (grape juice, fermentation volume, temperature) are the same in all the studies. In both F1-hybrids, the meiotic segregation yields few clones consuming up to 60% of malic acid (Figure 1b,c). Thanks to this first approach, we were able to select two progeny clones GS-28B from SBxGN offspring and FM-8D from M2xF15 one that had high *MAC* value of 58.0%and 45.4%, respectively. These high values correspond to the reported limits of *Saccharomyces cerevisiae* physiology [43,55,56,57]. 

To pursue trait improvement, strains GS-28B and FM-D8 were mated by manually paring spores of both strains. The resulting F1-hybrid (FMGS-1) should have inherited from both genetic backgrounds independent genetic factors controlling MAC that can be recombined through meiosis. Therefore, FMGS-1 was sporulated and a high number of spores were purified by taking advantage of the hydrophobicity properties of spore wall. A spore preparation of 95% was obtained and around 1000 clones were isolated on solid YPD and stored at −80 °C. Fifty randomly picked individuals were fermented in the M15 grape juice and their *MAC* was measured. In this population, the consumption levels of malic acid were on average significantly much higher than those previously observed in SBxGN and M2xF15 progenies as well as in the commercial starters panel (Figure 1d). This result demonstrates that the simple cross of two optimal individuals derived from different background is very efficient for optimizing complex phenotypes. Such strategy is routinely used by microbiologists for improving yeast strains and have been reported and reviewed by many authors [11,14,36,58,59]. 

### 3.2. Use of Marker Assisted Selection (MAS) for Enhancing Malic Acid Consumption 

To go further, we used at a large-scale genetic markers for selecting optimal individual prior any phenotypic tests. Indeed, getting genotypic data is generally easier than setting up a phenotypic characterization. For testing the relevance of this strategy, we first defined a set of eleven genetic makers that are nucleotide variations genetically linked to QTL controlling MAC and identified in SBxGN background [46,47]. The Figure 2a,b summarize SNP position and *enhancer* or *preserver* alleles that were define for each QTL. The genotype of the founder strains (SB, GN, M2 and F15) and the parental strains GS-28B and FM-8D is given panel c. We applied this genotypic screening on the ~1000 meiotic spores previously isolated from FMGS-1 and sought for meiotic spore clones having preferably inherited Quantitative Trait Loci (QTL) allele promoting the malic acid consumption. As detailed in the material and method section, each QTL was tracked by using the MASS-array technology able to discriminate in a unique experiment eleven pairs of biallelic Single Nucleotide Polymorphisms (SNP) (Appendix A). The inheritance of the 11 loci for 924 FMGS-1 progenies was interrogated. SNP calling was higher than 94% and most of the strains were fully genotyped. A first segregation analysis confirmed that spore purification was very efficient for removing residual F1-hybrids (fully heterozygous). Indeed, most of the clones (77%) were fully homozygous while a minor fraction (15.5%) harbored both homozygous and heterozygous loci and could be considered as sibling-pair hybrids (F2) (Appendix A). Only fully homozygous strains with less than one missing genotype were analyzed representing 583 individuals. According to parental genotypes, the segregation of five loci was expected (Figure 2). Indeed, 5 *enhancer* alleles were fixed (IV_360G, IV_414T, VII_480G, XI_382T, and XI_631T), one *preserver* allele was fixed (II_661G), and the five other loci segregated in a mendelian fashion (chi^2^test *p* > 0.05) (Figure 3a).

Next, we investigated the phenotypic impact of markers segregation. The progeny population was split in two groups according to the number of *enhancer* alleles detected. 122 individuals harbor 9 or 10 *enhancer* alleles while 125 individuals harbor only 5 or 6 *enhancer* alleles (Figure 3b). Each group represents nearly 20% of the total population genotyped. In order to test QTLs effect in the FMGS-1 background, 30 and 23 individuals belonging to each group were randomly selected and phenotyped in the Merlot grape juice. Strains enriched for *enhancer* alleles (9 to 10 optimal loci) consumed more malic acid than the control group (Figure 3c). On average, the *enhancer* group consumes 53.7% of malic acid while the control only 48.0% of malic acid (Wilcoxon test *p* < 0.05). Although the phenotypic gain observed (+5.7% of malic acid) was moderated, this demonstrates that MAS allows the blind selection of highly performing individuals without any phenotypic screening.

### 3.3. Construction and Characterization of a Backcrossed FMGS-2 Population

A possible cause of the moderate efficiency of *MAS* in the FMGS-1 background would be the fact that many QTLs were fixed (i.e., have a homozygous inheritance). Therefore, a second hybrid (FMGS-2) was obtained in order to increase the number of segregating loci. A progeny clone of FMGS-1 (FMGS-1#647) was crossed with GS-41B, one segregant of SBxGN hybrid. FMGS-1#647 has an optimal MAC (65.7%) and harbor 8 out 11 *enhancer* alleles. GS-41B has a moderated *MAC* value (40.4%) but allows the introduction of other enhancer alleles in the resulting hybrid. Indeed, in the FMGS-2 hybrid, all but two QTLs (VII_480 and XII_53) were heterozygous (Figure 4a). From FMGS-2, 1232 spore clones were isolated and genotyped using the same procedure than for FMGS-1 offspring. Mass-array analysis confirmed the Mendelian segregation of 9 out the 11 markers typed (chi^2^test *p* > 0.05). The genotyping of these clones allows the identification of nearly 12% of residual F1-hybrids and F2-sibling pair hybrids due to a less efficient spore purification. In the further analysis, 797 fully homozygous meiotic segregants were used. In order to test *MAS* efficiency in this new background, the same approach than for FMGS-1 was used. Fifty randomly selected individuals were phenotyped for characterizing the overall performance of FMGS-2 progeny. Random progenies of FMGS-1 and 2 have similar average values (52.6 and 48.7) (Table 2 and Appendix A), however the phenotypic variability (variance) in the FMGS-2 offspring is two folds higher (130 vs. 58). This result is likely due to the higher number of QTLs in segregation in FMGS-2 population. Beside this difference, the proportion of clones with optimal malic acid consumption is broadly the same (Table 2). 

Similarly, a *MAS* was carried out by selecting individuals according to their genotype. Seventeen individuals that have inherited of 10 and 11 *enhancer* loci were selected among the 797 FMGS-2 progeny genotyped. This *enhancer*-rich group was compared to a small group of 10 strains harboring only 2 to 4 *enhancers* loci. On average, the *enhancer* group consumes +8% of malic acid at the end of the alcoholic fermentation (53.7% vs. 45.8%, Wilcoxon test α = 0.05) (Figure 4c). This also demonstrates that, in the FMGS-2 background, *MAS* is a convenient way for selecting extreme individuals in a small group of strains. Indeed, within the 17 genetically selected strains, two strains consumed more than 70% of malic acid which represents 11% of the population (Table 2); this proportion is significantly higher than those expected in the random population (hyper geometric test α = 0.05). 

### 3.4. Study of QTL Penetrance in FMGS-1 and FMGS-2 Populations

Although a significant difference was found between *enhancers* and *preservers* groups (Figure 3c and Figure 4c), many individuals with a nearly optimal genotype reach a moderate MAC value. Encompassing the random and marker-selected strains of each FMGS populations, a total of 154 strains were phenotypically and genetically characterized (Appendix A). This set of strains was used for estimating the cumulative impact *enhancer* allele on the malic acid consumption. A linear model confirms that, the more the number of *enhancer* alleles, the stronger the percentage of malic acid consumed (Figure 5a). Although, this correlation is very significative (Spearman test, rho = 0.23, *p*-value < 0.0001); a great residual variation within extreme groups is still observed. Indeed, the number of QTLs explained only 4.2% of the total variance observed in the linear model (LM1 see methods). This result illustrates the low penetrance level of the QTL pool used, which is a frequent case in *MAS* programs [40]. 

The effect of each QTL on malic acid consumption was tested and compared to their effect observed in the SBxGN background that was used for their identification. When a significative difference was observed (one-way ANOVA α = 0.05), the part of variance explained was indicated (Table 3). Since all the loci did not segregate in both progenies this test was in some cases not applied.

Among the eleven QTLs used, only six have a significative effect in the SBxGN population (Anova α = 0.05). This result is explained by the fact that those QTLs were detected in three different environments [46] which is not the case in the present analysis of variance. Surprisingly, although the effect of the causative genes *PNC1* has been molecularly validated [46], the inheritance of the marker *VII_427* did not impact *MAC* in the population. When the two FMGSs populations were pooled, two of the four segregating QTLs had a significative effect (*IV_31* and *XV_1052*). The locus *IV_31* has a complete penetrance since is significative in the four populations. The locus *XV_1052* has a more contrasted effect; it accounted for 9.7% of total variance observed in FMGS-1 progeny but did not significantly impact the malic acid consumption in FMGS-2 or SBxGN populations. Finally, the locus *XI_631* has a significative effect only in the FMGS2 population (Table 3). These results show that most of the QTLs tracked have an incomplete penetrance in the FMGS hybrids. This result can be explained by at least three causes: (i) the limited number of progenies tested (less than 100), (ii) the epistatic relations withing QTLs, (iii) the presence of modifier loci. We tested possible pair-wise interactions within the nine segregating QTLs in the FMGS-2 population (Appendix A). Interestingly, two significative interactions were found (*XV_1052* vs. *XI_382*) and (*II_661* vs. *XV_491*). In the first case, the inheritance of the allele *XI_382^C^* impairs the enhancer effect of the allele *XV_1052^G^* (Figure 5b). This explains why *XV_1052^G^* allele has a much higher penetrance in the FMGS-1 progeny where the *XI_382^T^* allele is fixed (Figure 2). Concerning the *II_661* and *XV_491* interaction, the best combination expected *II_661^G^*/*XV_491^T^* has a similar effect than the worst one *II_661^T^*/*XV_491^C^* (Figure 5c); suggesting a possible metabolic trade-off between their associated genes *PYC2* and *PTC5*. 

### 3.5. Phenotypic Characterization of High Malic Acid Consuming Strains

Four malic consumer strains belonging to both FMGS’ populations (FMGS_215, FMGS_889, FMGS2_107, FMGS2_265) were selected in order to be phenotypically characterized in various environmental conditions. These strains were compared to SB and GN which represent the extreme limits of the starter population (Figure 1a). Small-volume alcoholic fermentations were carried out in two grape musts (M15 and SB16) adjusted with different concentrations of malic acid and nitrogen and incubated at two different temperatures (see methods). The objective was to assess the impact of several basic environmental factors in enology that could affect *MAC*. The part of variance of each factor as well as their first-order interactions were estimated by applying a fractional experimental design (see methods). The effect of each factor is represented in the Figure 6a and summarized in Appendix A. 

All conditions combined; *MAC* varies mainly with the nature of the must (55.2% of total variance explained). Indeed, yeast strains consumed in average 32.5% more malic acid in white must compared to red must (Figure 6a). In contrast, temperature had a low impact on MAC and can be considered as poorly significant in the present study (Figure 6b). The important must-effect observed is likely due to the highest malic acid concentrations found in white must (SB16) compared to those of the red one (M15). This difference strongly impacts the pH of the matrix and may affect the diffusion of malic acid through cell membrane since this acid is protonated below pH = 3.4 [60]. 

In order to dissociate this matrix effect, a second set of ANOVA was carried out by splitting the dataset according to the must. The new models shed light on the effect of nitrogen and malic acid concentrations that were partially hidden by the must effect. Nitrogen concentrations impacted malic acid consumption in both musts (Figure 6c). In our experiment, malic acid consumed was 10% higher in low nitrogen condition (150 mg/L N). However, this trend was not observed for the strain GN that consumes more malic acid (+4%) in rich nitrogen condition (300 mg/L N). The impact of grape juice concentration of malic acid on *MAC* was also strongly contrasted. As shown on the Figure 6d, the malic acid level accounts for 11.8% of the total variance explained in the white wine but only 0.9% of the total variance in the red one. In the white grape juice, the *MAC* decreases (−8%) for the highest malic acid concentration (9.39 g/L) while this trait is steadier in red grape juice. Again, the response of GN differs from the malic consumer strains. Indeed, in red must, the impact of initial malic acid was stronger, with an increased MAC with higher initial malic acid concentration.

Finally, this experiment demonstrates that in both musts, the factor “strain” has the major impact explaining 74.8% and 60.5% of “red” and “white” models, respectively (Figure 6e). As expected, all the strains selected by genetic markers consumed significantly more malic acid than the control GN. The strongest consumer is the strain FMGS_889 that metabolized 32% and 11% more malic acid than the strains GN and SB, respectively (Wilcoxon test *p* < 0.01). Although minor variations were attributed to nitrogen and malic acid content, those strains steady consume malic acid which is not the case of the strain GN. The founder strain SB belongs to the malic consuming group but has always a lower MAC than the four individuals selected. As a direct consequence, the resulting pH of wine at the end of the alcoholic fermentation was drastically impacted (Appendix A). Indeed, in the most discriminating conditions, the pH difference measured in produced wine between extreme strains (GN and FMGS_889) was of 0.42 units (M15_high_initial_24), demonstrating the strong impact of *S cerevisiae* strains in the modulation of wine pH.

### 3.6. Ongoing of Malolactic Fermentation

During the winemaking process, most of the red wines and some white wines underwent the malolactic fermentation (MLF) due to the development of the lactic bacteria *Oenococcus oeni* [44]. The malic acid is one key energetic substrate of the MLF and its content in wine at the end of the alcoholic fermentation has an obvious impact on the process. It has been reported that L-malic acid concentration delays the MLF duration but does not drastically affect the *O. oeni* fermentation rate [61,62]. In addition, low pH has a negative impact on MLF initiation and efficiency [44]. Since strains selected clearly impacted malic acid concentration (Figure 6e) and wine pH (Appendix A), their influence on MLF was evaluated by inoculating the different wines produced with a malolactic starter (see methods). MLF ongoing was monitored by measuring the L-malic acid degradation by an enzymatic assay. MLF kinetics were fitted by a logistic function and four kinetics parameters were extracted *(t_MLF50, t_MLFend, t_MLF_lag, rmax_MFL*) (Appendix A). For the malic consuming yeast strains, the MLFs were completed between 10 and 15 days. In contrast, MLFs carried out in wines fermented by GN were much slower and were sometimes not achieved (Figure 7a).

A correlation analysis performed on the 36 wines (6 strains and 6 conditions) clearly demonstrated a positive correlation between the initial malic acid content and the time for consuming 50% of malic acid (t_MLF50) (Figure 7b). Indeed, the amount of malic acid in wines fermented with GN was 1.1 g·L^−1^ higher, on average, than those from other strains (Figure 7c). GN was different from the other strains (Kruskal–Wallis test, α = 0.05) and MLFs of wines fermented with GN required twice more time for consuming 50% of total malic acid. In contrast, the maximum fermentation rate (*rmax_MLF*) was quite similar between the strains. Indeed, strains FMGS_889 and GN have similar *rmax_MLF* but opposed *t_MLF50* values (Figure 7d). Thus, the time for completing MLF (*t_MLFend*) proved to be faster (less than 300 h) when malic acid concentration was below 2 g·L^−1^. In addition, a low initial concentration in malic acid does not seem to impair the initiation of MLF since the lag time of MLF (*t_MLFlag*) is quite similar in all the conditions tested. Altogether, these results suggested that malic consuming strains are compatible with sequential inoculation of lactic acid bacteria and should facilitate the MLF by increasing wine pH and reducing the malic acid concentration to consume.

## 4. Discussion

### 4.1. Assesment of a Wide MAS Program for Improving a Complex Trait

Marker assisted selection (MAS) constitutes the applied side of quantitative genetics and has been widely used for improving plant varieties and animal races [32,63,64]. For the last two decades, *Saccharomyces cerevisiae* has been raised as a perfect model for addressing fundamental questions in quantitative genetics [65,66,67,68]. Furthermore, the budding yeast is also the prime industrial microorganism involved in bioethanol and fermented foods production. Although methods for identifying QTLs are perfectly controlled [31], their use for improving yeast strains for industrial applications is poorly documented. In the present study, we aimed to increase the malic acid consumption of wine starters in order to propose new strains with outlier features. To achieve this program, we get a new F1-hybrid (FMGS-1) resulting from the cross of two optimal unrelated strains (GS-28b and FM-8d). Then, we implemented for the first time a wide MAS program using 11 QTLs controlling the same trait.

First, a robust protocol was developed for isolating and genotyping a wide number of meiotic progenies derived from prototrophic and homothallic yeast. Although these features complexify breeding operations [14], the purification of FMGSs hybrids yielded more than 80% of spore clones (doubled haploids) or sibling-pair hybrids (F2) (Appendix A). The use of robotic-assisted DNA extraction and mass array genotyping allow a SNP call average of 94% and most of the strains were unambiguously genotyped. This technique allows the parallel genotyping of up to 40 markers which would strongly decrease the genotyping cost per SNP.

Second, we demonstrated that individuals carrying a high proportion of *enhancer* alleles consumes statistically more malic acid than those carrying a high proportion of *preserver* alleles in both FMGS progenies (Figure 3c and Figure 4c). This result confirms that a blind preselection of strains can be achieved on the basis of their genotype. Nevertheless, the overall effect of this pool of alleles is limited regards to the general cross effect observed. Indeed, many FMGS-1 and FMGS-2 progenies reach high *MAC* value whatever their genotype. This result indicates that the simple selection of optimal outbred parental strains is very efficient for optimizing quantitative phenotypes. Indeed, the average malic acid consumed by randomly selected FMGS-1 progenies (50 strains) is statistically similar to average value of the FMGS-1 *enhancer* group (52.6% vs. 53.7%). This is explained by the fact that five *enhancer* alleles are homozygous (fixed) in FMGS-1 hybrid and only five QTLs are in segregation (Figure 2a) reducing the proportion of non-optimal genotypes. Thus, 36% of randomly selected segregants (18) carry at least 8 *enhancer* alleles of the 11 possible one. These strains include FMGS-647, FMGS-215 and FMGS-889 that are among the top malic consumers. A second blind selection was then carried out in the FMGS-2 progeny. Since 10 of 11 QTL segregated, the difference of MAC between *enhancer* and *preserver* groups was larger (8%). A tight significative difference was found between random and *enhancer* enriched groups (48.7% vs. 53.7%, respectively, Wilcoxon test < 0.1). In addition, the proportion of optimal clones consuming more than 65 or 70% is higher than in random population (Table 2). This highest proportion supports the idea that MAS would be efficient for the blind selection of many optimal individuals and should be used as a prescreening step. Since many other quantitative traits segregate independently in FMGS’s offspring, MAS may allow the selection of a collection of top-degrading malic acid strains to be used in downstream selection programs focusing on other traits (aromatic, fermentation performances, temperature adaptation…). 

### 4.2. Possible Causes of Incomplete QTLs Penetrance

Although it was successfully applied in many organisms, the efficiency of MAS is strongly depending on the number of marker used, the type of breeding design, the complexity, and the number of traits to improve [32,40,64,69]. Indeed, when QTLs detected in a specific background are introduced in an outbred individual, their expressivities are much lower than those observed in the initial QTL population [40]. This general trend was also observed here since only three QTLs of the eleven used has a significant effect of FMGSs hybrids (Table 2). This incomplete penetrance can be explained by several causes. First, the 11 QTLs used were detected in a multiple environmental design which increase the detection power of QTLs [46,47]. Thus, it is not surprising that minor QTLs showing GxE does not raise the significant threshold imposed in the subset of 94 SBxGN progenies. Second, possible alleles controlling *MAC%* in the M2xF15 background were not considered at all. Since 50% of FMGS-1 genome is constituted by M2 or F15 alleles, QTLs governing the phenotypic variability of M2xF15 background are likely influencing the *MAC%* of FMGS’s progeny. Third, the lack of penetrance observed for the marker VII_427 related to the causative gene *PNC1* would result to the complex genetic structure of the chromosome VII that contains many causative genes including *PMA1* a major contributor of MAC% [43]. Finally, some QTLs seems to show epistatic interactions (Figure 5b,c) that modulate their expressivity in a non-predicable manner. 

### 4.3. Outlier Strains for Specific Enological Applications. Lowering the Acidity of Rich Malic Wines and Shortening the MLF of Red Wines

From an enological point of view, this study provides new strains for managing atypical vinification conditions. Indeed, *S. cerevisiae* strains consuming more than 50% of the total malic acid are quite rare and have been scarcely described [55,57,70,71]. Such strains would be useful to better managing wine acidity and may have two distinct enological applications. First, in very acidic grape juice (cold climate white grapes), the excess malic acid is a concern since the taste of related wines is quite sour [72]. Therefore, the biotechnological reduction of malic acid would contribute to provide smoother wines (less aggressive) from a tasty point of view. In the Sauvignon blanc grape juice containing 9.39 g/L of L-malic acid, the difference in final L-malic acid concentration reached 1.3 g/L (Figure 7a) representing a pH shift close to 0.2 pH unit (Appendix A) with a possible impact on the acidity perception. Second, such strains would contribute to facilitate malolactic fermentation by increasing wine pH and reducing L-malic concentration. Indeed, we demonstrated that none of the FMGSs strains selected has a negative impact MLF ongoing inoculated with a commercial starter (Lactoenos^®^ B7 Direct, Laffort oenology, FRANCE). In contrast, the malic acid reduction and the pH increase promote the MLF by reducing its duration. Therefore, whatever the residual amount of L-malic acid at the end of the alcoholic fermentation, the MLF is initiated readily without any significative impact on the consumption rate of bacteria. 

By measuring the *MAC%* in several musts and conditions (temperature, nitrogen composition and L-malic concentration) we demonstrated that the high consumption properties of FMGS’s strains are quite robust to environmental changes. Temperature did not impact this trait and the main factor is the nature of grape juice (Figure 6a). Although the compositions of the two grape juices are not chemically defined, a possible cause of this discrepancy should be the pH of the media that affects the charge and the transport of malic acid [70]. However, in a specific must, L-malic acid concentration has moderated effect (Figure 6c). Beside the nature of the must, the impact of nitrogen seems to be the most important factor with strain specific interactions that has been previously reported [57]. Indeed, malic-consumer strains achieved a stronger MAC in nitrogen rich conditions which is not the case of GN. The GN’s behavior was previously reported for other wine yeast strains [57]. This suggest that the high malic-consumer strains selected would have a particular response to nitrogen level.

Although significantly improved, the consumption of the total amount of malic acid by *S. cerevisiae* seems to be illusive. Indeed, the strain FMGS_889 consumed 84% of malic acid in SB16 and 67% in M15. In both cases, the final concentration of malic acid fall to 0.9 g/L constituting a kind of physiological threshold. This limit is likely due to the absence of an active malic acid transporter able to pump in this organic acid through the cell membrane [73]. A further improvement of malic acid consuming ability would require the enhancement of malic acid uptake. Beside the expression of recombined proteins [41], this new skill might be acquired by activating quiescent transmembrane transporter by mutagenesis as achieved for xylulose import [74]. 

## Figures and Tables

**Figure 1 jof-07-00304-f001:**
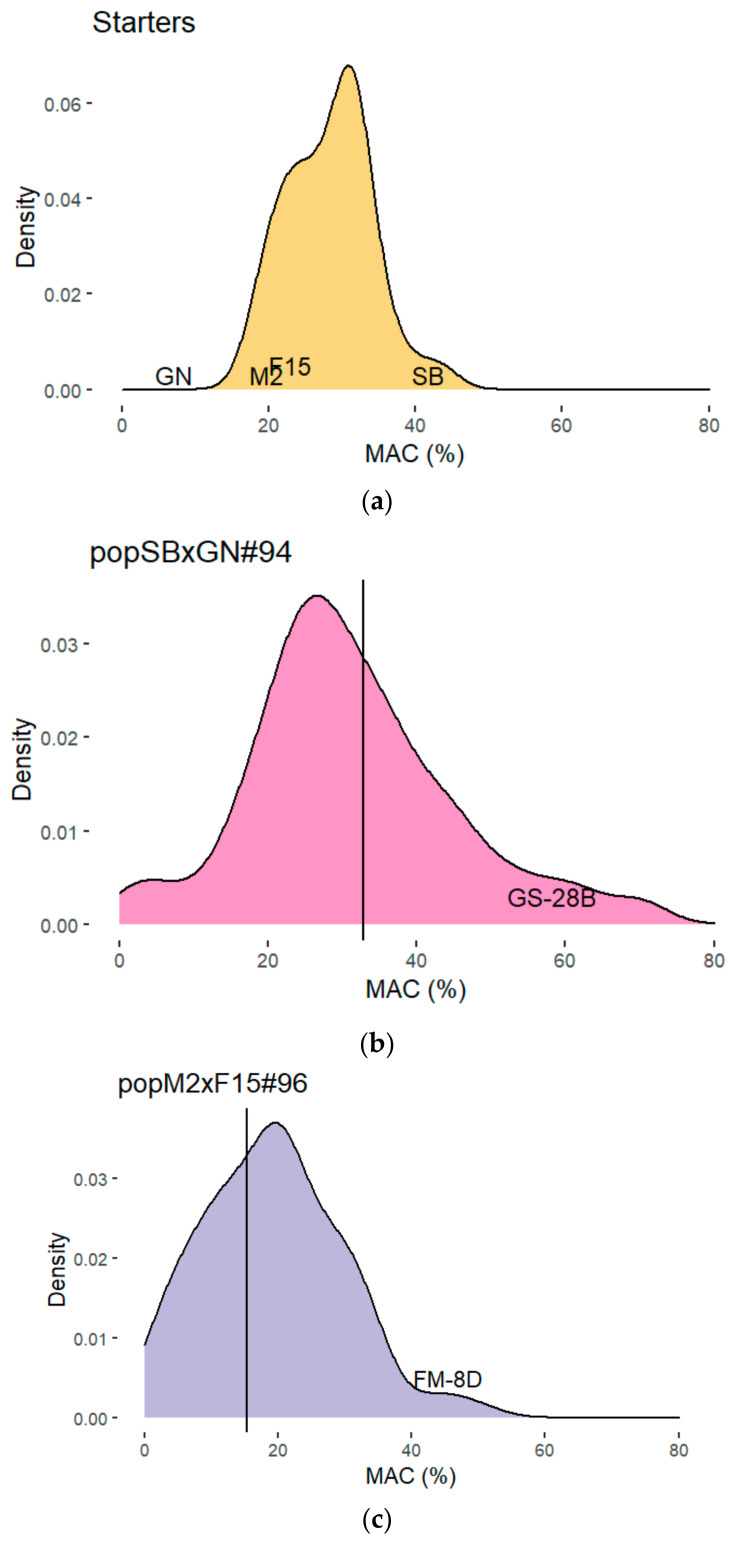
Malic acid consumed (MAC) at the end of alcoholic fermentation in a merlot grape juice. (**a**) Phenotypic distribution of 31 starters and relative position of the 4 founder strains SB, GN, M2, F15 (data previously published by [52]). (**b**) Phenotypic distribution of SBxGN hybrid and 94 related progeny clones including GS-28B (data previously published [46]). (**c**) Phenotypic distribution of M2xF15 hybrid and 96 related progeny clones including FM-8D (this work). All fermentations were carried out in the same conditions. (**d**) Phenotypic distribution of FMGS1 hybrid and 50 related progeny clones. The vertical bar indicated the value of the F1-hybrid of each progeny.

**Figure 2 jof-07-00304-f002:**
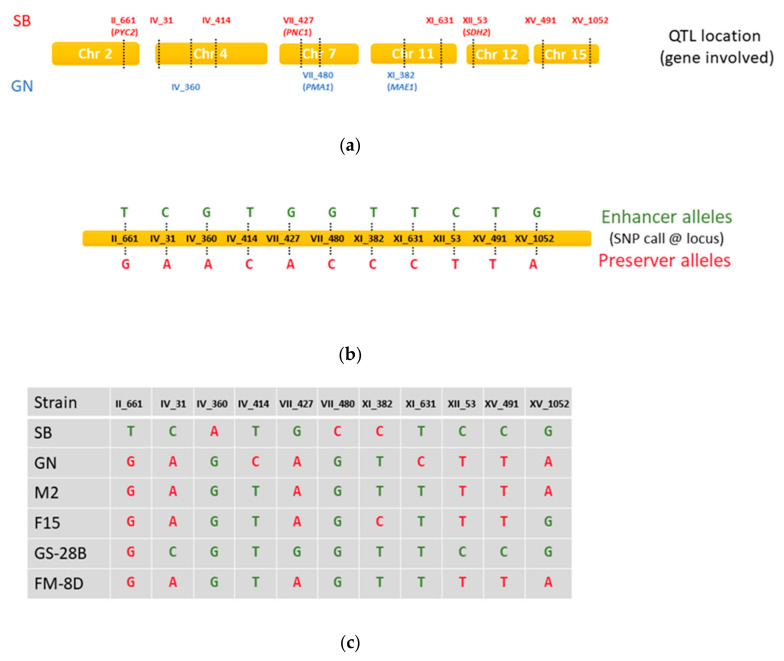
Quantitative trait loci (QTLs) tracked, and mass-array markers used, dotted lines represent the position of a marker. (**a**) Genetic position of the eleven QTLs mapped modulating malic acid consumption in SBxGN hybrid. The parental allele (GN or SB inheritance) promoting a stronger consumption was indicated for each locus. (**b**) SNP corresponding to the enhancer and preserver allele, the exact position is given in Appendix A. (**c**) SNP call in the four founder and in the parental strains GS-28B and FM-8D.

**Figure 3 jof-07-00304-f003:**
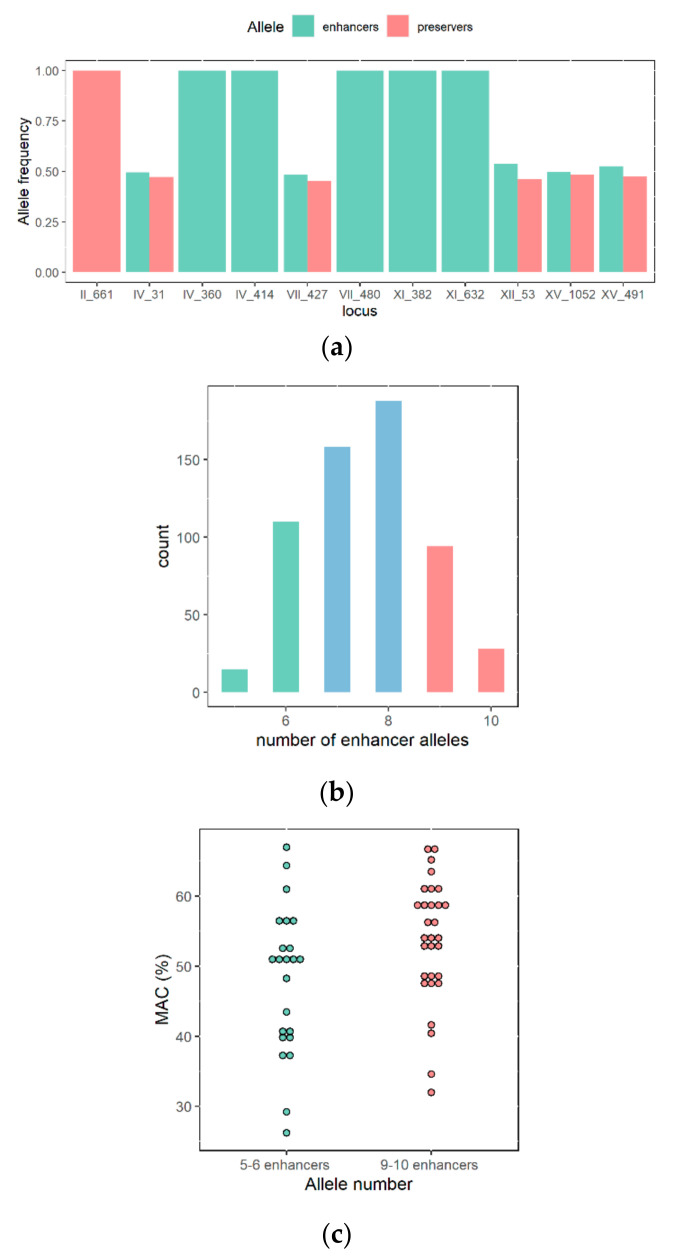
Marker assisted selection of FMGS-1 progeny for five segregating alleles. (**a**) Allele frequency in the progeny, enhancer and preserver alleles are shown in green and red, respectively. (**b**) Distribution of the number enhancer alleles counted in the FMGS-1 progeny (593 fully genotyped progenies). (**c**) Phenotypic difference between progenies having enriched and minimized number of enhancer alleles; values indicated represented the average percentage of Malic acid Consumed (MAC) for each strain (3 replicates).

**Figure 4 jof-07-00304-f004:**
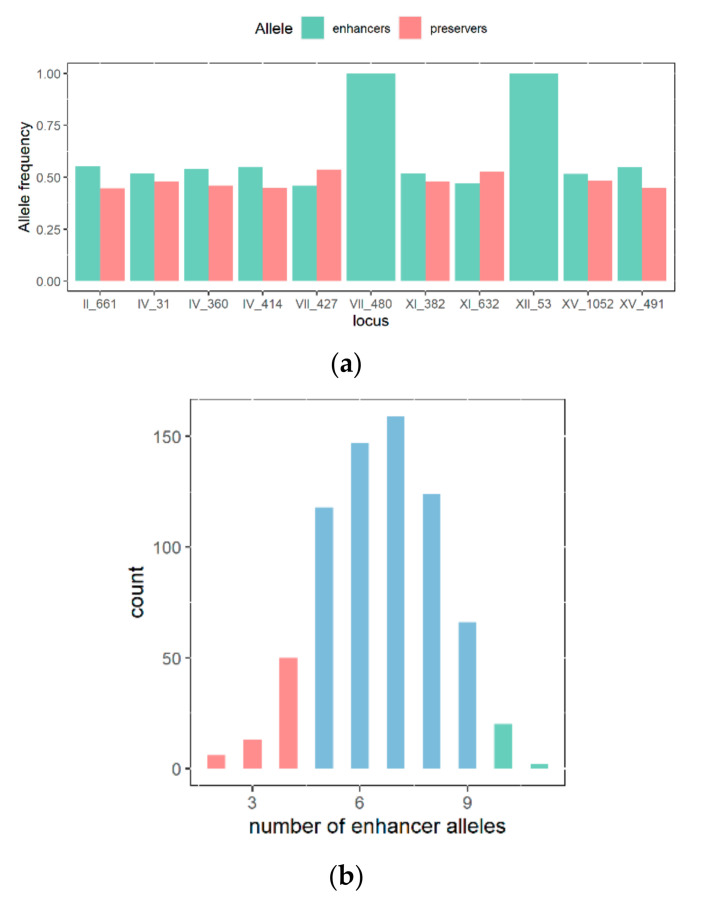
Marker assisted selection of FMGS-2 progeny for nine segregating alleles. (**a**) Allele frequency in the progeny, enhancer and preserver alleles are shown in green and red, respectively. (**b**) Distribution of the number enhancer alleles counted in the FMGS-2 progeny (797 fully genotyped progenies). (**c**) Phenotypic difference between progenies having enriched and minimized number of enhancer alleles; values indicated represented the average percentage of Malic acid Consumed (MAC) for each strain (3 replicates).

**Figure 5 jof-07-00304-f005:**
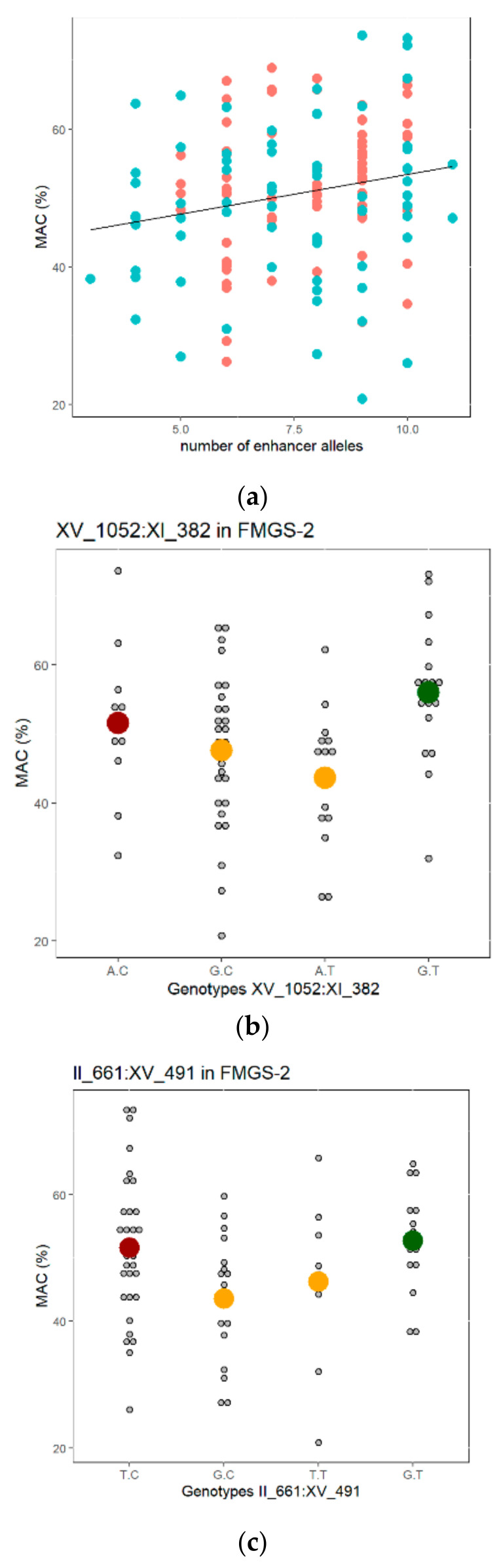
Overall QTL effect and pair wise interactions. (**a**) Linear relationship between the number of enhancer alleles and the MAC value in the 154 FMGS’ progenies tested; the red and blue dots represent the FMGS-1 and FMGS-2 progenies, respectively. (**b**,**c**) Genetic interactions between QTLs observed in the FMGS-2 population (72 individuals). The red, orange, and green dots are the average values observed for the bad, intermediate, and good combinations expected.

**Figure 6 jof-07-00304-f006:**
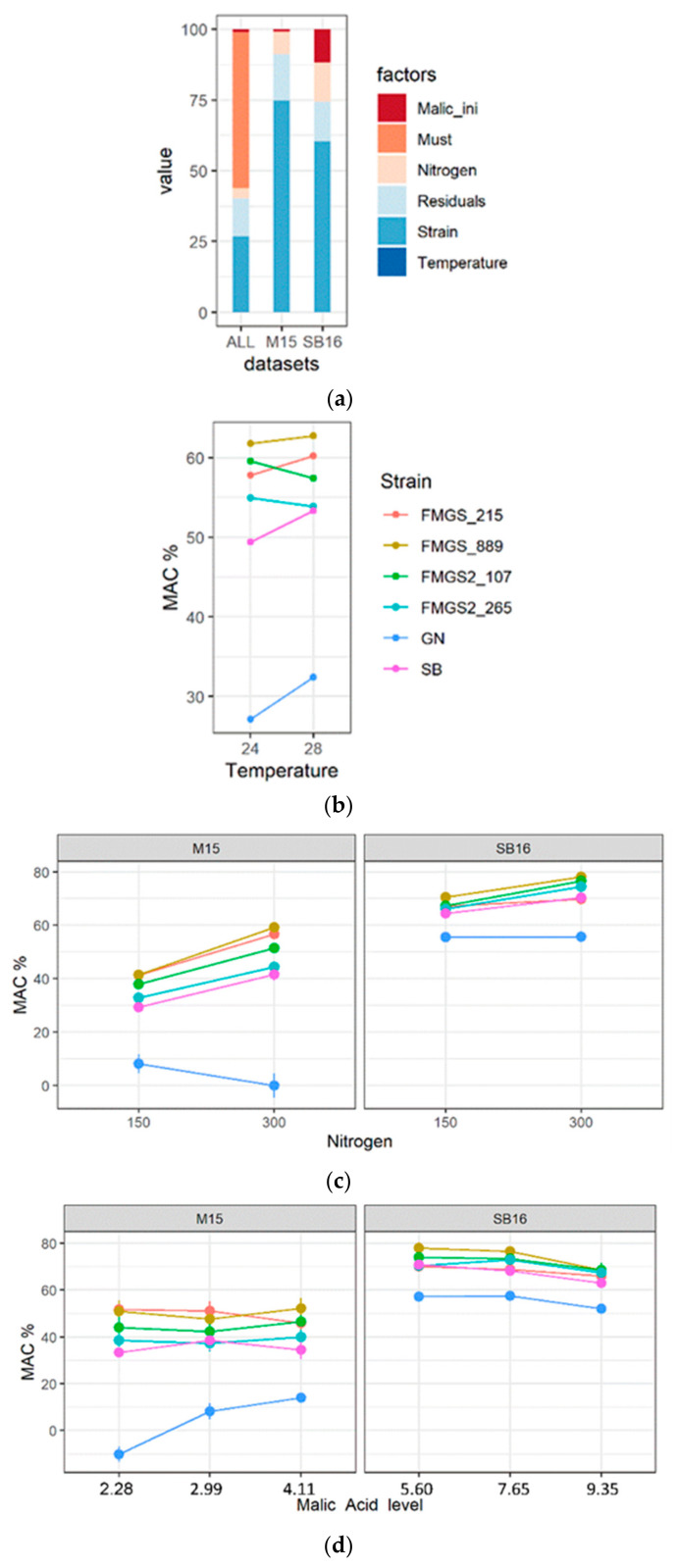
Biotic and abiotic factors affecting the variation of malic acid consumption. (**a**) Bar graphs indicate the part of variance explained by the different factors considered by the linear model applied: malic acid concentration, must origin, nitrogen content, strain origin, and temperature. (**b**) Temperature effect according to the strains used; (**c**) nitrogen effect according to the strains used; (**d**) initial malic acid concentration effect according to the strains used; (**e**) strain effect in both grape juices. For panels (**b**–**e**) the full dots represent the average values.

**Figure 7 jof-07-00304-f007:**
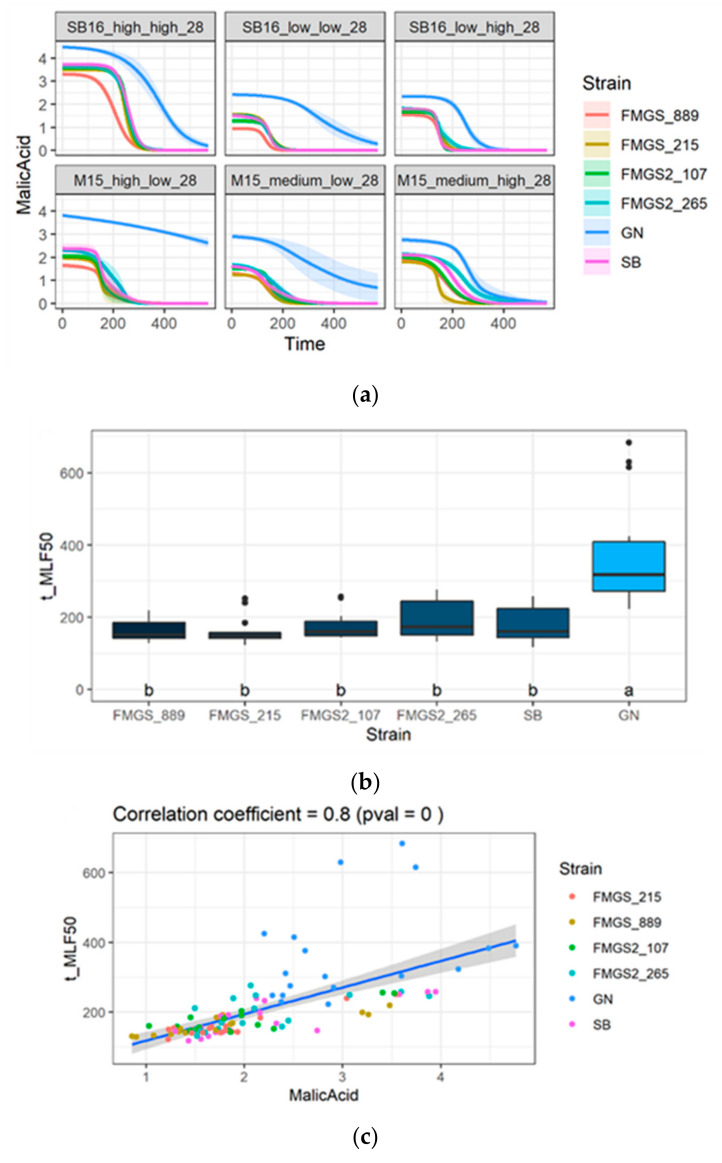
Malolactic fermentations with high malic consumer strains. (**a**) Malic acid concentration (g/L) according to the time (h) for each stain used in the 6 inoculated wines. (**b**) Time (h) to achieve 50% of the MLF according to the strains used. (**c**) Correlation between the time at 50% of the MLF (t_MLF50) and the malic acid concentration (g/L). (**d**) Maximum fermentation rate (g/L/h) according to the strains used.

**Table 1 jof-07-00304-t001:** Main *Saccharomyces cerevisiae* strains used.

Strain	Description	Reference	WDCM-791 (CRB ISVV, Bordeaux)
M2	parental strain, a meiotic spore clone from Enoferm M2 (Lallemand, Canada)	[26]	CRBO L2010
F15	parental strain, a meiotic spore clone from Zymaflore F15 (Laffort, France)	[26]	CRBO L2011
GN	parental strain, a meiotic spore clone from Zymaflore VL1 (Laffort, France)	[48]	CRBO L2002
SB	parental strain, a meiotic spore clone from Zymaflore BO213 (Laffort, France)	[48]	CRBO L2001
M2xF15	F1-hybrid (M2 x F15)	[26]	CRBO L2005
SBxGN	F1-hybrid (SB x GN)	[48]	CRBO L2003
GS-28b	meiotic clone of SBxGN	this work	CRBO L2012
GS-41b	meiotic clone of SBxGN	this work	CRBO L2013
FM-8d	meiotic clone of M2xF15	this work	CRBO L2014
FMGS-1	F1-hybrid (GS-28b x FM-8d)	this work	CRBO L2015
FMGS-1-647	meiotic clone of FMGS-1, malic acid consumer strain	this work	CRBO L2016
FMGS-2	F1-hybrid (GS-41b x FMGS-1-647)	this work	CRBO L2017
FMGS-1-889	meiotic clone of FMGS-1, malic acid consumer strain	this work	CRBO L2018
FMGS-1-215	meiotic clone of FMGS-1, Malic acid consumer strain	this work	CRBO L2019
FMGS-2-107	meiotic clone of FMGS-2, Malic acid consumer strain	this work	CRBO L2020
FMGS-2-265	meiotic clone of FMGS-2, Malic acid consumer strain	this work	CRBO L2021

**Table 2 jof-07-00304-t002:** Biometric comparison of the populations used.

Populations	Number	Mean	Variance	Quantiles			% of Individuals with a MAC Above	
				50%	75%	95%	65%	70%
Starters	31	28.3	33.3	28.2	31.5	36.5	no	no
pop SBxGN	93	31.0	206.6	28.6	36.9	58.5	2.2	2.2
pop M2xF15	94	17.8	138.9	18.6	24.8	34.1	no	no
random FMGS-1	50	52.6	58.6	51.7	57.2	65.7	12.0	no
selected FMGS-1 ^a^	30	53.7	77.0	54.1	59.2	65.8	10	no
random FMGS2-1	50	48.7	130.4	49.0	56.6	64.3	4.0	2.0
selected FMGS2-2 ^b^	17	53.7	121.2	54.2	57.1	72.3	17.6	11.7

^a^ Genetically selected for 9–10 enhancer alleles; ^b^ genetically selected for 10–11 enhancer alleles; no: not observed.

**Table 3 jof-07-00304-t003:** QTL penetrance in various populations ^a.^

Population	II_661	IV_31	IV_360	IV_414	VII_427	VII_480	XI_382	XI_631	XII_53	XV_491	XV_1052
Gene validated	*PYC2*				*PNC1*	*PMA1*	*MAE1*		*SDH2*	*PTC5*	
SBxGN (*n* = 94)	3.87	4.22	ns	ns	ns	8.20	2.97	ns	4.45	7.61	ns
FMGSs (*n* = 154)	-	7.14	-	-	ns	-	-	-	-	ns	4.92
FMGS-1 (*n* = 82)	-	9.31	-	-	ns	-	-	-	ns	ns	9.74
FMGS-2 (*n* = 72)	ns	6.41	ns	ns	ns	-	ns	3.10	-	ns	ns

^a^ Part of variance explained in %, one-way analysis of variance, *p*-value <0.05. ns: not significative, -: not relevant.

## Data Availability

Strains used have been deposited on the WDCM-791 (CRB ISVV, Bordeaux).

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
