# Peer review of "Marker Assisted Selection of Malic-Consuming Saccharomyces cerevisiae Strains for Winemaking. Efficiency and Limits of a QTL’s Driven Breeding Program"

_jof, 2021, doi:10.3390/jof7040304_

Round 1
Reviewer 1 Report
The work is up-to-date, I take up the key topics for the wine industry, among others. It covers the aspect of genetic selection of strains, which is very important, especially in the last decade of climate changes and their impact on the vinification process. Moreover, the use of selection (MAS) substantially supports the selection of the appropriate strain.
The work was properly planned. The methodology was correctly selected, adequate to the research scope. The obtained research results were clearly and legibly presented by means of graphs and statistically analyzed. The discussion of the results also does not raise any objections. Only according to the reviewer, the manuscript lacks a summary. I think it should be there.
Author Response
we thanks the reviewer for its kind comments
Reviewer 2 Report
To increase the malic acid consumption of wine starters in order to propose new strains, the authors performed to get a new F1-hybrid (FMGS-1) resulting from the cross of two optimal unrelated strains (GS-28b and FM-8d) and implemented for the first time a wide MAS program using 11 QTLs controlling the same trait. Moreover, the author demonstrated that the high consumption properties of FMGS’s strains are quite robust to environmental changes. These results would be meaningful for gaining basic knowledge academically and developing of brewing breeding technology in this report. Accordingly, I consider this MS as acceptable to be published in Journal of Fungi. However, there are some revision before publication. Additionally, check the English through manuscript.
Minor comments:
P3L97 Table 1
A part of Description in Strain (FMGS-2)
“GS-41b x FMGS-647” to “GS-41b x FMGS-1-647”?
P8L304
“six” to “five”?
Fig. 3c and Fig. 4c
Describe the value of average.
P10L367 Table 2
Please correct the above statement “% of strains with MAC higher than” as it is difficult to understand.
P11L428 Fig. 5c
“XII_661” to “II_661”
P12L465
“(300 mg/L N)..” to “(300 mg/L N).”
Author Response
We thanks the reviewer for its comments and we clarified each points highlighted. In addition we corrected some typos and rechecked the grammar of the manuscript carrefully.
Minor comments:
P3L97 Table 1
A part of Description in Strain (FMGS-2)
“GS-41b x FMGS-647” to “GS-41b x FMGS-1-647”?
yes corrected
P8L304
“six” to “five”?
exact we changed here and in two lines before
Fig. 3c and Fig. 4c
Describe the value of average.
we clarified this point in both captions
P10L367 Table 2
Please correct the above statement “% of strains with MAC higher than” as it is difficult to understand.
we clarified this point in the table 2
P11L428 Fig. 5c
“XII_661” to “II_661”
corrected
P12L465
“(300 mg/L N)..” to “(300 mg/L N).”
corrected